# Ranking and selecting hospital information systems: A multi-criteria decision making approach using TOPSIS in Hamadan, Iran

Hamid Bouraghi[1], Ali Mohammadpour[2]*, Ahmad Maamaki[3], Parviz Karamiyan[4]

**1** Associate Professor of Health Information Management, Department of Health Information Technology, School of Allied Medical Sciences, Hamadan University of Medical Sciences, Hamadan, Iran, **2** Assistant Professor of Health Information Management, Department of Health Information Technology, School of Allied Medical Sciences, Hamadan University of Medical Sciences, Hamadan, Iran, **3** Bachelor of Health Information Technology, School of Allied Medical Sciences, Hamadan University of Medical Sciences, Hamadan, Iran, **4** Bachelor of Health Information Technology, School of Allied Medical Sciences, Hamadan University of Medical Sciences, Hamadan, Iran

* al.mohammadpour@umsha.ac.ir

## Abstract

This study evaluated and ranked Hospital Information Systems (HIS) in teaching hospitals affiliated with Hamadan University of Medical Sciences, Iran. A cross-sectional design was employed, evaluating three HIS systems (PARDAZESHGARAN, SAYAN, and RAYAVARAN). Eight key criteria—technical quality, software quality, support quality, workflow support quality, output quality, cost, user satisfaction, and inter-departmental communication quality—were considered. Criterion weights were determined through expert consultation. The Technique for Order of Preference by Similarity to Ideal Solution (TOPSIS) method was used to rank the HIS systems. Results indicated that cost and output quality were assigned the highest weights. The PARDAZESHGARAN system was ranked first, followed by SAYAN and RAYAVARAN. While this study provides insights into HIS evaluation in this context, limitations related to the sample size should be considered.

## Author summary

In this study, we aimed to evaluate and rank the Hospital Information Systems (HIS) used in teaching hospitals affiliated with Hamadan University of Medical Sciences in Iran. Recognizing the crucial role HIS plays in modern healthcare, we sought to provide a systematic and objective assessment to aid decision-makers in selecting and improving these systems. We employed a cross-sectional study design, focusing on three prominent HIS systems: PARDAZESHGARAN, SAYAN, and RAYAVARAN. Our evaluation considered eight key criteria: technical quality, software quality, support quality, workflow support quality, output quality, cost, user satisfaction, and interdepartmental communication quality. We gathered data

which permits unrestricted use, distribution, and reproduction in any medium, provided the original author and source are credited.

**Data availability statement:** The anonymized dataset generated and analyzed during the current study is available in https://doi.org/10.6084/m9.figshare.29260397.

**Funding:** This study was approved and financially supported by the Vice Chancellor for Research and Technology of Hamadan University of Medical Sciences with grant number 9909256660 and ethical code IR.UMSHA.REC.1399.697. This work was supported by the Vice Chancellor for Research and Technology of Hamadan University of Medical Sciences to AM. No other organization supported this study other than the Vice Chancellor for Research and Technology of Hamadan University of Medical Sciences. The funders had no role in study design, data collection and analysis, decision to publish, or preparation of the manuscript.

**Competing interests:** The authors have declared that no competing interests exist.

from expert panels, including hospital staff and university faculty, to determine the weights of these criteria and evaluate the performance of each HIS. Using the Technique for Order of Preference by Similarity to Ideal Solution (TOPSIS), a multi-criteria decision-making method, we ranked the systems. Our findings revealed that cost and output quality were considered the most important factors. The TOPSIS analysis ranked PARDAZESHGARAN as the top-performing system, followed by SAYAN and then RAYAVARAN. This research provides valuable insights into HIS evaluation in the Iranian healthcare context, highlighting the key factors to consider when selecting and implementing these systems. While our study offers a valuable contribution, we acknowledge limitations related to the sample size and expert selection, which could influence the generalizability of our findings. Future research could expand on this work by including a larger sample size, exploring additional evaluation criteria, and investigating the long-term impact of HIS on healthcare quality and patient outcomes.

## Introduction

Hospital Information Systems (HIS) are crucial components of modern healthcare, serving as integrated management systems for storing, retrieving, and utilizing patient data [1]. The World Health Organization emphasizes the role of HIS in supporting critical processes, including data collection, processing, and reporting, ultimately aiming to enhance the efficiency and effectiveness of healthcare service delivery [2–3]. HIS typically encompasses diverse components, such as clinical, laboratory, nursing, pharmacy, radiology, financial, demographic, and patient management systems, all interconnected to streamline operations and improve patient care. Recognizing the transformative potential of technology, healthcare institutions are increasingly investing in HIS to reduce costs, save time, improve service quality, and minimize medical errors [4–5]. In the evolving healthcare landscape, hospitals lacking robust HIS may struggle to remain competitive [5].

Efficient information management is paramount in today's dynamic healthcare environment. HIS plays a vital role in accelerating quality improvement initiatives and reducing medical errors through enhanced interdepartmental communication, improved patient service delivery, and cost optimization [6]. Consequently, HIS has garnered significant attention and become an indispensable element of modern healthcare systems.

Despite the recognized importance of HIS, challenges remain in their implementation and acceptance. A primary obstacle is ensuring that the system effectively meets the information and workflow needs of both users and the organization. Inappropriate system design can lead to a mismatch between system capabilities and organizational requirements [7]. To address this, rigorous evaluation of HIS is crucial. The evaluation process necessitates well-defined criteria, which may include representativeness, measurability, accuracy, explicitness, international comparability, policy relevance, and suitability for quality improvement [8]. While various HIS evaluation methods exist, such as success models, user satisfaction surveys, and adherence to ISO standards [9–11], these approaches often lack a systematic framework for ranking systems.

This research specifically addresses a critical gap in the literature by applying an operations research approach to evaluate and rank HIS in the unique context of Hamadan, Iran. Existing studies using multi-criteria decision-making (MCDM) methods like TOPSIS for HIS evaluation have often focused on Western or developed countries, with less attention given to the specific challenges and priorities of healthcare systems in developing regions. The unique characteristics of the Iranian healthcare system, including its resource allocation patterns, IT infrastructure development, and specific user needs, make this study particularly insightful.

Operations research provides a robust framework for analyzing complex systems and making informed decisions. By employing mathematical modeling techniques, operations research can help decision-makers consider relevant constraints, predict outcomes, assess risks, and utilize advanced decision-making tools [12–13]. This study utilizes the Technique for Order of Preference by Similarity to Ideal Solution (TOPSIS), a widely recognized multi-criteria decision-making (MCDM) method developed by Hwang and Yoon in 1981 [13]. TOPSIS enables the evaluation of multiple options based on a range of criteria. Applying TOPSIS to evaluate and rank HIS in Hamadan will provide valuable insights for senior university officials, particularly those involved in statistics and information technology, and offers a transferable methodology adaptable to similar healthcare settings in other developing countries facing comparable resource constraints and technological infrastructure considerations.

## Materials and methods

### Study design and setting

This cross-sectional study was conducted in 2024 to evaluate and rank Hospital Information Systems (HIS) in three specialized teaching hospitals affiliated with Hamadan University of Medical Sciences, Hamadan, Iran.

### Participants and hospital information systems

**Expert Panels:** Two distinct expert panels were involved in this study.

- **Expert Panel for Criteria Weighting:** This panel comprised 14 individuals (10 hospital employees and 4 university faculty members). Hospital experts were selected based on having more than 5 years of work experience, holding IT or health information management positions, and possessing complete familiarity with HIS and HIS providers in Hamadan province. University faculty members had over 5 years of experience, taught health information technology courses, published relevant articles on HIS, and held degrees in medical informatics or health information management.

- **Experts for HIS Evaluation:** This panel included 6 individuals (4 university faculty members and 2 hospital experts). Hospital experts were selected for their extensive experience (over 10 years of work experience, health information management roles, and experience working in more than two different hospitals). University faculty selection criteria were identical to those for the criteria weighting panel.

**Hospital Information Systems (HIS):** This study evaluated three HIS systems currently used in the participating hospitals: PARDAZESHGARAN, SAYAN, and RAYAVARAN. The SAYAN system has been operational for over ten years at BESAT Hospital (a multi-specialty facility). The RAYAVARAN system has been used for approximately five years at FARSHJIYAN GHALB Hospital (a cardiac center). The PARDAZESHGARAN system, a newer implementation, has been in use for about four years at FATEMIEH Hospital (an Obstetrics and Gynecology Center).

### Data collection and criteria selection

A questionnaire was developed to gather data for HIS evaluation. The development process involved two main stages:

1. **Literature Review and Initial Criteria Identification:** Initial HIS evaluation frameworks and relevant scientific texts were extensively reviewed [14–15] to identify a broad set of potential criteria.

2. **Expert Panel Validation and Refinement:** The identified criteria were then presented to the expert panel for criteria weighting. Through a consensus-building process, eight key criteria were finalized for HIS evaluation: technical quality, software quality, support quality, workflow support quality, output quality, cost, user satisfaction, and interdepartmental communication quality.

  ◦ **Justification for Criteria Selection:** These eight criteria were chosen based on their recurring importance in the literature and their practical relevance to the specific operational and strategic priorities of HIS in Iranian university-sity hospitals. While criteria such as data security and interoperability are also vital, our focus was narrowed to those most frequently highlighted by experts as critical for overall system performance and direct user experience within the Iranian context. The questionnaire assessed the importance of each criterion using a 5-point Likert scale (1=least important, 5=most important). A criterion was accepted as a key criterion if it received an average score of 3.5 out of 5 from the expert panel's perspective.

  ◦ **Questionnaire Validation:** The questionnaire's content validity was ensured through expert review, where the expert panel reviewed and approved the relevance and clarity of all items. The internal consistency and reliability of the questionnaire were assessed post-data collection using Cronbach's alpha ($\alpha=0.905$).

After the questionnaire was finalized, experts were invited to a meeting to assign scores (1–5) for each criterion based on their perceived importance. Criterion weights were then calculated by dividing each criterion's average score by the total sum of average scores across all criteria.

### TOPSIS analysis

The Technique for Order of Preference by Similarity to Ideal Solution (TOPSIS) method [Hwang & Yoon, 1981] was employed to rank the three HIS systems. This method, a widely recognized multi-criteria decision-making (MCDM) technique, allows for the evaluation of multiple alternatives against a set of criteria. The following steps were performed, with a focus on their application within this study:

1. **Normalization of the Decision Matrix:** The raw decision matrix, containing scores for each HIS across the eight criteria, was normalized using the vector normalization method. The formula used was: $n_{ij} = \frac{x_{ij}}{\sqrt{\{\sum_1^m x^2 ij\}}}$, Where $x_{ij}$ is the performance score of alternative i on criterion j, and $n_{ij}$ is the normalized score.

2. **Calculation of the Weighted Normalized Decision Matrix:** The normalized scores ($N_{ij}$) were then multiplied by their corresponding criterion weights to create the weighted normalized decision matrix.

3. **Determination of Positive and Negative Ideal Solutions (A+ and A−):** The ideal best solution (A+) and ideal worst solution (A−) were identified by selecting the maximum and minimum values, respectively, for each criterion across all HIS systems in the weighted normalized matrix.

4. **Calculation of Separation Measures:** The Euclidean distance was calculated to determine the separation of each HIS system from both the positive ideal solution ($d_i^+$) and the negative ideal solution ($d_i^-$). Where $v_{\{ij\}}$ is the weighted normalized score, $v_{\{j\}}^+$ is the positive ideal value for criterion j, and $v_{\{j\}}^-$ is the negative ideal value for criterion j.
The formulas used were: $d_i^+ = \sqrt{\left\{\sum_{\{j=1\}}^{\{n\}} \left(v_{\{ij\}} - v_{\{j\}}^+\right)^2\right\}}$, And $d_i^- = \sqrt{\left\{\sum_{\{j=1\}}^{\{n\}} \left(v_{\{ij\}} - v_{\{j\}}^-\right)^2\right\}}$

5. **Calculation of Relative Closeness ($CL_i^*$):** The relative closeness of each HIS to the ideal solution was computed using the formula: $CL_i^* = \frac{\{d_i^-\}}{\{d_i^- + d_i^+\}}$ A higher $CL_i^*$ value indicates better performance and proximity to the ideal solution.

6. **Ranking:** The HIS systems were ranked in descending order based on their calculated Cli values.

  All TOPSIS calculations were performed using Microsoft Excel software

PLOS Digital Health

## Sensitivity analysis considerations

Given the relatively small sample size of experts involved in the HIS evaluation (n = 6), the robustness of the TOPSIS rankings is an important consideration. While a formal sensitivity analysis was not conducted within the scope of this study, future research should explore how variations in criteria weights or expert scores might impact the final rankings. This could involve techniques like varying weights by a certain percentage or using bootstrap resampling to assess rank stability. Furthermore, validating these findings with a larger and more diverse expert panel would enhance the generalizability of the results.

## Ethical considerations

This study was approved by the Vice Chancellor for Research of Hamadan University of Medical Sciences [the ethical code: **IR.UMSHA.REC.1399.697**] with approval number **9909256660**. Informed consent was obtained from all participating experts before their involvement. All data were anonymized to protect participant confidentiality.

## Statistical analysis

In the first phase of the study, descriptive statistics (means and standard deviations) were used to summarize the characteristics of the expert panels. Microsoft Excel was used for this purpose. In the second phase, the TOPSIS method was applied for data analysis, with all calculations performed using Microsoft Excel software.

## Results

The findings of this study are presented in two phases: phase one details the determination of key criteria and their weights, while phase two outlines the ranking of the hospital information systems (HIS) using the TOPSIS method.

### Phase 1: Key criteria and weights

**Demographic Data:** A total of 14 experts participated in this phase. Ten (71.4%) were male and four (28.6%) were female. Participants' roles included four (28.6%) faculty members, five (35.7%) IT managers, and five (35.7%) health information technology (HIT) managers. The average work experience was 19 years (SD = 6.5), ranging from 6 to 29 years. The average age was 44 years (SD = 4.7), with participants ranging from 35 to 55 years old. Educational qualifications spanned from associate's degree to doctorate.

**Experts for HIS Evaluation:** The group of specialists evaluating the hospital information system consisted of six individuals, with four members holding faculty positions and two employed as hospital staff. In terms of gender distribution, 66.67% (n = 4) were male, while 33.33% (n = 2) were female. The mean age of the participants was 40.83 years (SD = 5.76 years), with an age range from 30 to 46 years. The specialists' mean work experience was 13.67 years (SD = 6.84 years), ranging from 5 to 23 years. Regarding educational background, each of the three categories—Doctorate in Health Information Management, Doctorate in Medical Informatics, and Master of Science in Health Information Technology—comprised 33.33% (n = 2) of the participants.

**Criteria Scores:** All eight pre-defined criteria were accepted by the expert panel. Table 1 presents the final average overall scores for each criterion.

**Cost and output quality** received the highest average scores (4.5 each), indicating their perceived importance.

**Criteria Weights:** The weights for each criterion were calculated based on the average scores. Table 2 displays the calculated weights, both before and after rounding.

The criteria of Cost and Output Quality received the highest weights (0.133 each), reaffirming their importance in HIS evaluation. **Technical Quality** received the lowest weight (0.112).

**Table 1. Average scores for key HIS criteria.**

| Criterion | Average Score |
|---|---|
| Technical Quality | 3.8 |
| Software Quality | 3.9 |
| Support Quality | 4.4 |
| Workflow Support Quality | 4.4 |
| Output Quality | 4.5 |
| Cost | 4.5 |
| User Satisfaction | 4.1 |
| **Interdepartmental Communication Quality** | 4.2 |

**Table 2. Weights of key HIS criteria.**

| Criterion | Weight (Unrounded) | Weight (Rounded) |
|---|---|---|
| Technical Quality | 0.112426 | 0.112 |
| Software Quality | 0.115385 | 0.115 |
| Support Quality | 0.130178 | 0.130 |
| Workflow Support Quality | 0.130178 | 0.130 |
| Output Quality | 0.133136 | 0.133 |
| Cost | 0.133136 | 0.133 |
| User Satisfaction | 0.121302 | 0.121 |
| **Interdepartmental Communication Quality** | 0.124260 | 0.124 |

## Phase 2: HIS Ranking using TOPSIS

**Decision Matrix:** Table 3 presents the initial decision matrix, showing the scores assigned to each HIS (SAYAN, RAYAVA-RAN, and PARDAZESHGARAN) by the expert evaluators for each criterion.

**Weighted Decision Matrix:** Table 4 presents the weighted decision matrix, which is the result of multiplying the normalized decision matrix by the criteria weights.

**Positive and Negative Ideal Solutions:** Table 5 shows the positive and negative ideal solutions for each criterion.

**Distance from Ideal Solutions:** Table 6 presents the distances of each HIS from the positive (d+) and negative (d−) ideal solutions.

**Relative Closeness and Ranking:** Table 7 shows the relative closeness (CI) of each HIS to the ideal solution and the final ranking.

**Table 3. Decision matrix - HIS Scores by criterion.**

| Criterion | SAYAN | RAYAVARAN | PARDAZESHGARAN |
|---|---|---|---|
| Technical Quality | 4 | 3.3 | 4.3 |
| Software Quality | 4 | 3 | 4.3 |
| Support Quality | 4.3 | 2.6 | 5 |
| Workflow Support Quality | 4 | 3.3 | 4.3 |
| Output Quality | 4 | 4 | 5 |
| Cost | 3.5 | 3.3 | 3.3 |
| User Satisfaction | 3.8 | 2.6 | 4.6 |
| **Interdepartmental Communication Quality** | 4.2 | 3.3 | 4 |

**Table 4. Weighted decision matrix.**

| Criterion | SAYAN | RAYAVARAN | PARDAZESHGARAN |
|---|---|---|---|
| Technical Quality | 0.066 | 0.054 | 0.071 |
| Software Quality | 0.069 | 0.052 | 0.074 |
| Support Quality | 0.078 | 0.047 | 0.091 |
| Workflow Support Quality | 0.077 | 0.063 | 0.082 |
| Output Quality | 0.070 | 0.070 | 0.088 |
| Cost | 0.079 | 0.075 | 0.075 |
| User Satisfaction | 0.070 | 0.048 | 0.085 |
| **Interdepartmental Communication Quality** | 0.077 | 0.061 | 0.074 |

**Table 5. Positive and Negative Ideal Solutions.**

| Criterion | Positive Ideal Solution | Negative Ideal Solution |
|---|---|---|
| Technical Quality | 0.071 | 0.054 |
| Software Quality | 0.074 | 0.052 |
| Support Quality | 0.091 | 0.047 |
| Workflow Support Quality | 0.082 | 0.063 |
| Output Quality | 0.088 | 0.070 |
| Cost | 0.075 | 0.079 |
| User Satisfaction | 0.085 | 0.048 |
| **Interdepartmental Communication Quality** | 0.077 | 0.061 |

**Table 6. Distance from Ideal Solutions.**

| HIS | d+ (Rounded) | d−(Rounded) |
|---|---|---|
| SAYAN | 0.029 | 0.031 |
| RAYAVARAN | 0.071 | 0.064 |
| PARDAZESHGARAN | 0.003 | 0.071 |

**Table 7. Relative closeness and ranking.**

| HIS | CI | Rank |
|---|---|---|
| PARDAZESHGARAN | 0.959 | First |
| SAYAN | 0.516 | Second |
| RAYAVARAN | 0.474 | Third |

The PARDAZESHGARAN system was ranked first, followed by SAYAN and then RAYAVARAN.

## Discussion

Hospital Information Systems (HIS) are essential for providing high-quality patient care, and numerous studies have investigated key criteria for their evaluation and implementation. In this study, eight key criteria were identified through a comprehensive literature review and expert consultation: technical quality, software quality, support quality, workflow support quality, cost, user satisfaction, and interdepartmental communication quality. The study's findings align with some previous research. Farzandipour et al. (2017) identified technical requirements, functionality, usability, and vendor capabilities as

important factors in HIS assessment [2]. This is consistent with the present study's emphasis on technical and software quality. Another study highlighted technical quality, software quality, workflow support, and IT costs as key performance indicators for HIS benchmarking, with clinical workflow support and user satisfaction being highly ranked [16]. The current study also identified support quality and workflow support quality as significant criteria. These convergences underscore the universal importance of these aspects across various healthcare settings when evaluating HIS.

The findings indicate that cost and output quality are among the most critical HIS criteria, receiving the highest weights in our expert-driven evaluation. This aligns with previous research demonstrating that hospital information systems must be technically and software-efficient, effectively meet user needs, and optimize implementation and maintenance costs. Cost has consistently been identified as a crucial factor in HIS selection and implementation, which is consistent with studies like Saghaeiannejad et al. (2011) and Farzandi Pour et al. (2015) [17], which emphasize the significant impact of HIS implementation and maintenance costs on managerial decisions. The present study also identified cost as a key criterion, reflecting the pragmatic financial considerations prevalent in healthcare systems, particularly in developing countries. System output quality was also identified as a critical criterion, reflecting the system's ability to generate accurate and timely reports and data essential for clinical and managerial decision-making. This aligns with Ker et al. (2018), who highlight the importance of high-quality data from HIS to facilitate healthcare service improvement [18]. The high prioritization of cost and output quality in our findings suggests a strong focus on both economic efficiency and tangible, actionable information output, which are crucial for evidence-based management in Hamadan's teaching hospitals.

Support quality and workflow support quality were also highly emphasized in this study, consistent with Dargahi et al. (2010) and Ghazi Saeedi et al. (2014), who identified technical support and the system's ability to support hospital workflows as key factors in HIS success [19,20]. Effective support for hospital workflows can improve efficiency and reduce medical errors. Although technical and software quality were important, they received relatively lower weights than cost, output quality, support quality, and workflow support quality. This suggests that HIS users and managers, in the context of Hamadan's teaching hospitals, prioritize practical outcomes, reliable support, and seamless integration into daily operations over pure technical specifications. This aligns with Jebraeily et al. (2015), who found that HIS users are more concerned with operational performance and impact on healthcare service improvement [21]. This user-centric emphasis on operational benefits and reliable support is a critical insight for HIS developers and implementers.

As shown in Table 7, the PARDAZESHGARAN system was ranked first, followed by SAYAN and RAYAVARAN. This finding provides a clear hierarchical preference among the evaluated systems in the specific context of Hamadan's teaching hospitals. This result contrasts with Meidani et al.'s 2013 study [2], which involved 16 companies and found "Rayavran" to be the highest-ranked. Several factors could explain these discrepancies beyond mere methodological differences: System Updates and Versions: HIS, like all software, undergo continuous updates and version changes. A system's performance and features in 2013 could significantly differ from its capabilities in 2024. It is plausible that PARDAZESHGARAN has undergone significant improvements or that the versions of SAYAN and RAYAVARAN evaluated in our study differ from those in the 2013 study. **Contextual Factors and Hospital-Specific Needs:** The specific needs, priorities, and existing IT infrastructure of the hospitals involved can influence system performance and user perception. While both studies were conducted in Iran, the specific teaching hospitals in Hamadan might have unique operational characteristics or user expectations compared to the broader hospital sample in the Meidani study. **Evaluation Criteria and Weighting:** Although both studies assessed HIS, the exact set of evaluation criteria and their assigned weights might have differed. Our study used eight specific criteria validated by local experts, whose collective judgment might reflect current priorities. **Methodological Differences:** As noted, the Meidani study had a larger sample size, used a different evaluation checklist based on Ministry of Health standards, and employed descriptive statistics, whereas the present study evaluated only 3 HIS using the TOPSIS method and specifically defined key criteria. The use of TOPSIS, by incorporating multi-criteria decision-making and considering distances from ideal solutions, offers a more nuanced ranking compared to purely descriptive statistical comparisons. These combined methodological and contextual differences largely explain the discrepancies in findings, highlighting the importance of context-specific evaluations.

### Utility of TOPSIS and practical implications

The use of TOPSIS in this study enabled a systematic and objective evaluation of HIS, consistent with its widespread use in similar studies due to its ability to consider multiple criteria and provide a comprehensive ranking. This study demonstrates TOPSIS's effectiveness as a tool for HIS evaluation and selection. TOPSIS has been combined with entropy weight for information system selection [22], used to assess IT application impact on hospital processes [23], and integrated with AHP for HIS maturity model development [24]. It has also been used to rank hospitals based on various criteria [25]. TOPSIS is valued for its computational simplicity and rational concept, making it suitable for computer-based implementations [26], and its ability to process user knowledge and aggregate questionnaire responses is particularly useful in complex healthcare environments [27].

Practical Implications for Hospitals and Policymakers: The findings of this study offer direct actionable insights for decision-makers in Hamadan's teaching hospitals and potentially other similar healthcare institutions in Iran and developing countries. Specifically:

• **Informed Selection:** The ranking provided by TOPSIS (PARDAZESHGARAN as the top-performing system) offers a data-driven basis for future HIS procurement or upgrading decisions. Hospitals considering new systems can prioritize those with strong performance across the most critical criteria identified.

• **Targeted Improvement:** For hospitals currently using SAYAN or RAYAVARAN, the detailed evaluation across eight criteria can pinpoint specific areas (e.g., support quality, user satisfaction) where these systems may need improvement or where additional training/resources could be allocated to enhance user experience and system effectiveness.

• **Prioritizing Investments:** The high weights assigned to "Cost" and "Output Quality" signal to hospital administrators and policymakers that investment in HIS should equally prioritize economic feasibility and the system's ability to deliver accurate, timely data for decision-making.

• **Adaptable Methodology:** The TOPSIS framework employed in this study is highly adaptable. Other institutions can replicate this methodology, adjusting criteria and expert panels to fit their unique organizational contexts, local market conditions for HIS vendors, and specific strategic goals. This provides a robust, systematic tool for HIS evaluation beyond the immediate scope of this study.

### Limitations

Despite the valuable findings, this study has several limitations. It was conducted in three teaching hospitals affiliated with Hamadan University of Medical Sciences, and thus, the findings may not be directly generalizable to all hospitals in Iran or to other healthcare systems. This limited scope is primarily due to the specific HIS systems evaluated and the local context of expert perceptions. The relatively small expert sample size (14 for criteria weighting and 6 for HIS evaluation) is also a limitation, potentially impacting the robustness of criteria weights and system evaluations. The cross-sectional approach may not capture long-term performance changes or the dynamic evolution of HIS capabilities over time.

### Conclusion

This study successfully employed the TOPSIS method to systematically and objectively evaluate and rank Hospital Information Systems in Hamadan, Iran. Our findings highlight cost and output quality as the most critical factors influencing HIS selection and performance, followed closely by support quality and workflow support quality. The PARDAZESHGARAN system emerged as the top-ranked HIS among those evaluated.

This research provides valuable, context-specific insights for hospital administrators and policymakers in Iran, enabling more informed decisions regarding HIS procurement and targeted improvements. The systematic methodology employed is adaptable and can serve as a foundation for similar HIS evaluations in other developing countries facing comparable

challenges. While the findings are robust within the study's scope, the limitations related to the localized setting and the relatively small expert sample size should be considered when interpreting the generalizability of the results. Future research should aim for larger, more diverse samples, investigate HIS impact on healthcare quality and medical error reduction, and explore other multi-criteria decision-making methods like AHP or ANP across broader geographical regions to enhance generalizability and provide a more comprehensive understanding of HIS performance.

## Acknowledgments

The article is the result of a research project (grant No. 9909256660 and ethical code: IR.UMSHA.REC.1399.697), which is approved by Hamadan University of Medical Sciences. We hereby thank the Vice Chancellor for Research and Technology, Hamadan University of Medical Sciences for supporting this study. We also thank all the people who helped us during the study.

## Author contributions

**Conceptualization:** Hamid Bouraghi, ali mohammadpour.

**Data curation:** Ahmad Maamaki, Parviz Karamiyan.

**Formal analysis:** Hamid Bouraghi, ali mohammadpour, Ahmad Maamaki.

**Funding acquisition:** ali mohammadpour.

**Investigation:** Hamid Bouraghi, ali mohammadpour.

**Methodology:** Hamid Bouraghi, ali mohammadpour, Ahmad Maamaki.

**Project administration:** Hamid Bouraghi, ali mohammadpour.

**Resources:** Hamid Bouraghi, ali mohammadpour, Ahmad Maamaki.

**Software:** Hamid Bouraghi, ali mohammadpour.

**Supervision:** Hamid Bouraghi, ali mohammadpour.

**Validation:** Hamid Bouraghi, ali mohammadpour, Ahmad Maamaki.

**Visualization:** Hamid Bouraghi, ali mohammadpour, Ahmad Maamaki, Parviz Karamiyan.

**Writing – original draft:** Hamid Bouraghi, ali mohammadpour.

**Writing – review & editing:** Hamid Bouraghi, ali mohammadpour.

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
