## [Decision Letter · Decision Letter 0]

20 May 2025

PDIG-D-25-00100Ranking and Selecting Hospital Information Systems: A Multi-Criteria Approach Using TOPSIS in Hamadan, IranPLOS Digital Health Dear Dr. mohammadpour, Thank you for submitting your manuscript to PLOS Digital Health. After careful consideration, we feel that it has merit but does not fully meet PLOS Digital Health's publication criteria as it currently stands. Therefore, we invite you to submit a revised version of the manuscript that addresses the points raised during the review process. Please submit your revised manuscript within 60 days Jul 19 2025 11:59PM. If you will need more time than this to complete your revisions, please reply to this message or contact the journal office at digitalhealth@plos.org. Please include the following items when submitting your revised manuscript: * A rebuttal letter that responds to each point raised by the editor and reviewer(s). You should upload this letter as a separate file labeled 'Response to Reviewers'. This file does not need to include responses to any formatting updates and technical items listed in the 'Journal Requirements' section below.* A marked-up copy of your manuscript that highlights changes made to the original version. You should upload this as a separate file labeled 'Revised Manuscript with Track Changes'.* An unmarked version of your revised paper without tracked changes. You should upload this as a separate file labeled 'Manuscript'. If you would like to make changes to your financial disclosure, competing interests statement, or data availability statement, please make these updates within the submission form at the time of resubmission. Guidelines for resubmitting your figure files are available below the reviewer comments at the end of this letter. We look forward to receiving your revised manuscript. Kind regards, Ali NabavizadehGuest EditorPLOS Digital Health Ali NabavizadehGuest EditorPLOS Digital Health Leo Anthony CeliEditor-in-ChiefPLOS Digital Healthorcid.org/0000-0001-6712-6626 **Journal Requirements:**1. Your current Financial Disclosure states, “We received 4 million tomans in Iranian currency for this research, equivalent to 45 US dollars.”. However, your funding information on the submission form indicates that you received funding from “Vice Chancellor for Research and Technology, Hamadan University of Medical Sciences”. Please indicate by return email the full and correct funding information for your study and confirm the order in which funding contributions should appear. Please be sure to indicate whether the funders played any role in the study design, data collection and analysis, decision to publish, or preparation of the manuscript. 2. Please provide a complete Data Availability Statement in the submission form, ensuring you include all necessary access information or a reason for why you are unable to make your data freely accessible. If your research concerns only data provided within your submission, please write "All data are in the manuscript and/or supporting information files" as your Data Availability Statement. 3. We have noticed that you have cited Table 9 in the manuscript file but there are no corresponding tables in the manuscript. Please amend your manuscript to include this table, noting that tables should not be uploaded as individual files. **Additional Editor Comments (if provided):**reviewer #1:

Overall Assessment

The manuscript is a Multi-Criteria Approach Using TOPSIS in Hamadan, Iran" (PDIG-D-25-00100) submitted to PLOS Digital Health. As a reviewer, I find the manuscript to be a valuable contribution to the field of health informatics, addressing a critical need for systematic evaluation of Hospital Information Systems (HIS) in the context of Iranian teaching hospitals. The use of the TOPSIS method to rank HIS based on multi-criteria decision-making is novel and relevant, aligning well with the journal’s focus on advancing digital health technologies. However, the manuscript has several significant issues that require substantial revisions to meet the standards of PLOS Digital Health. Based on the importance of the topic and the potential for improvement, I recommend acceptance with major revisions. Below, I outline the key strengths, issues, and specific revisions needed to ensure the manuscript’s suitability for publication.

The manuscript addresses a timely and relevant topic by evaluating HIS, a cornerstone of digital health, using a rigorous multi-criteria decision-making approach (TOPSIS). Its focus on Iranian teaching hospitals fills a gap in the literature, where systematic HIS evaluation studies are limited. The study’s methodology, including the use of expert panels and literature-based criteria (e.g., cost, output quality, support quality), is well-grounded and demonstrates scientific rigor. The clear structure, academic tone, and alignment with ethical standards (e.g., ethical approval, informed consent) further support its potential for publication. The findings, particularly the ranking of PARDAZESHGARAN as the top HIS and the emphasis on cost and output quality, provide actionable insights for healthcare administrators.

Major Issues

Despite its strengths, the manuscript has several critical issues that must be addressed to ensure scientific accuracy, transparency, and impact. These issues necessitate major revisions, as they involve methodological corrections, enhanced clarity, and deeper interpretation.

Methodological Errors in TOPSIS Analysis:

o A critical error exists in Table 5 (p. 15, p. 33), where the positive and negative ideal solutions for the cost criterion are reversed. As cost is a negative criterion (lower is better), the positive ideal should be the lowest value (0.075), and the negative ideal should be the highest (0.079). This error likely affects the distances (Table 6) and relative closeness (Table 7), potentially altering the final rankings. This undermines the study’s validity and requires recalculation.

o The TOPSIS methodology description in the methods section is incomplete, covering only normalization and weighted matrix steps. Key steps, such as calculating distances to ideal solutions and relative closeness (CI), are omitted, reducing reproducibility.

Limited Methodological Transparency:

o The questionnaire development process lacks detail on the literature review or frameworks used (e.g., DeLone and McLean model, ISO standards), limiting transparency.

o The small sample sizes (14 experts for criteria weighting, 6 for HIS evaluation, and three hospitals) are acknowledged as a limitation but not justified. A rationale citing practical constraints or similar studies is needed.

o The ethical considerations section is brief, lacking specifics on consent procedures (e.g., written or verbal) and data protection measures (e.g., anonymization protocols).

Clarity and Presentation Issues:

o The results section lacks a narrative explanation of the TOPSIS process, making it difficult for readers unfamiliar with MCDM to follow how scores lead to rankings.

o An incorrect reference to Table 9 (p. 17, line 287) instead of Table 7 in the discussion risks confusing readers and indicates a need for careful proofreading.

o The absence of figures (e.g., a bar chart or radar plot comparing HIS scores) limits the visual communication of results, which is a missed opportunity for a journal emphasizing clear data presentation.

Weaknesses in Discussion and Conclusion:

o The discussion overinterprets small differences in criteria weights (e.g., technical quality: 0.112 vs. user satisfaction: 0.121) as evidence that users prioritize practical outcomes, requiring a more nuanced interpretation.

o The discussion does not sufficiently explore discrepancies with prior studies or discuss practical implications for stakeholders, limiting the study’s impact.

o The conclusion is brief and does not articulate the broader applicability of the findings or include a call to action for stakeholders.

Minor Issues

Inconsistent Terminology:

o The manuscript uses “multi-criteria approach” in the title and abstract but occasionally refers to “multi-criteria decision-making” elsewhere (e.g., introduction, p. 7). Consistent use of “multi-criteria decision-making” (MCDM), the standard term in operations research, would improve professionalism.

o The HIS name “Rayavaran” is inconsistently capitalized (e.g., “RAYAVARAN” in Table 3, p. 32, vs. “Rayavaran” in the text, p. 16). This should be standardized throughout.

Formatting Inconsistencies in Tables:

o The decimal places in tables are inconsistent. For example, Table 2 (p. 31) reports weights to three decimal places (e.g., 0.133), while Table 5 (p. 33) uses three or four decimal places (e.g., 0.075 vs. 0.0792). All numerical values should use a consistent number of decimal places (e.g., three decimals throughout).

o Table titles and captions could be more descriptive. For instance, Table 1 (p. 31) is titled “Average Scores for Key HIS Criteria,” but it could specify “Based on Expert Panel Evaluation” to clarify the source.

Minor Proofreading Errors:

o There are minor grammatical errors, such as “hospitals lacking robust HIS may struggle to remain competitive” (p. 7, introduction), which could be rephrased for clarity (e.g., “Hospitals without robust HIS may face challenges in maintaining competitiveness”).

o The phrase “data are ready for delivery following the journal’s request” (p. 4) is awkward. A clearer phrasing would be “Data are available upon journal request” or “Data will be provided as per journal requirements.”

Lack of Footnotes in Tables:

o Tables 1–7 lack footnotes explaining key terms or methods, such as the 5-point Likert scale (Table 1), normalization process (Table 4), or ideal solutions (Table 5). Adding brief footnotes would enhance accessibility for readers unfamiliar with TOPSIS or the study’s methods.

Vague Wording in Abstract:

o The abstract’s mention of “sample size” as a limitation (p. 1) is vague. Specifying whether this refers to the number of hospitals, experts, or systems evaluated would improve clarity without requiring major revision.

o The term “expert consultation” in the abstract is imprecise. A minor clarification, such as “expert panel scoring,” would align with the methods section.

Underdeveloped Contextual Details:

o The introduction briefly mentions the Iranian healthcare context but lacks specific details, such as HIS adoption rates or challenges in teaching hospitals. Adding a sentence or two with relevant statistics or context would strengthen the rationale without requiring major restructuring.

o The demographic data of the expert panels (p. 12, p. 30) are presented but not integrated into the narrative. A minor addition explaining their relevance (e.g., “The panel’s expertise ensured robust criteria weighting”) would enhance clarity.

Limited Use of Abbreviations:

o The manuscript introduces “Hospital Information Systems (HIS)” but does not consistently use the abbreviation throughout (e.g., “hospital information systems” appears in full in the discussion, p. 17). Consistently using “HIS” after its first introduction would improve readability.

Minor Redundancy in Discussion:

o The discussion repeats the importance of cost and output quality (p. 17) without adding new insights in some sentences. Streamlining these points (e.g., combining redundant statements) would improve conciseness.

Reviewer #2:

1. Overview and Main Claims

The manuscript presents a study evaluating and ranking three Hospital Information Systems (HIS)—PARDAZESHGARAN, SAYAN, and RAYAVARAN—used in teaching hospitals affiliated with Hamadan University of Medical Sciences, Iran. The study employs the Technique for Order of Preference by Similarity to Ideal Solution (TOPSIS), a multi-criteria decision-making (MCDM) method, to rank the systems based on eight criteria: technical quality, software quality, support quality, workflow support quality, output quality, cost, user satisfaction, and interdepartmental communication quality. The main claims are:

• Cost and output quality are the most critical criteria for HIS evaluation.

• PARDAZESHGARAN ranks highest, followed by SAYAN and RAYAVARAN.

• The TOPSIS method provides a systematic and objective framework for HIS evaluation.

The study is relevant to the field of digital health, as HIS are integral to modern healthcare systems, and systematic evaluation methods can guide decision-making for system selection and improvement.

2. Evaluation Against PLOS Digital Health Criteria

Originality

The study applies TOPSIS, a well-established MCDM method, to evaluate HIS in a specific Iranian context, which adds some novelty to the application of operations research in healthcare informatics. However, the use of TOPSIS for HIS evaluation is not entirely novel, as referenced studies (e.g., Huang, 2008; Ma et al., 2022) have used TOPSIS in similar contexts. The manuscript acknowledges prior work but could better articulate how its specific focus on Hamadan’s teaching hospitals and the selected criteria contribute to the literature. For instance, the authors could highlight whether the criteria weighting or the local context introduces unique insights compared to previous studies.

Recommendation: Strengthen the originality claim by explicitly discussing how this study’s context, criteria selection, or findings differ from or complement existing TOPSIS-based HIS evaluations. If applicable, cite any unique challenges or characteristics of the Iranian healthcare system that make this study distinct.

Importance and Broad Interest

The topic is highly relevant to researchers, clinicians, and healthcare administrators in digital health, particularly in resource-constrained settings where HIS selection impacts cost, efficiency, and care quality. The emphasis on cost and output quality aligns with global priorities for optimizing healthcare technology investments. However, the study’s focus on three hospitals in Hamadan limits its immediate applicability to broader contexts. The findings could interest those working on HIS evaluation in similar settings, but the manuscript would benefit from discussing how the methodology or insights could be generalized or adapted elsewhere.

Recommendation: Expand the discussion to address how the methodology or findings could apply to other regions or healthcare systems. Highlight the broader implications for digital health stakeholders, such as policymakers or HIS vendors.

Methodological Rigor and Ethical Standards

The study employs a cross-sectional design with a clear methodology:

• Criteria Selection: Eight criteria were identified through literature review and validated by an expert panel, with a threshold of 3.5/5 for acceptance. The process is transparent, though the rationale for selecting these specific criteria over others (e.g., security, scalability) could be clearer.

• Expert Panels: Two panels (14 for criteria weighting, 6 for HIS evaluation) were carefully selected based on experience and expertise. The demographic data and selection criteria are well-documented, enhancing reproducibility.

• TOPSIS Application: The TOPSIS method is appropriately applied, with detailed steps (normalization, weighted decision matrix, ideal solutions, and relative closeness) and supporting tables. Excel was used for calculations, which is acceptable for this scale but limits scalability for larger datasets.

• Ethical Considerations: The study was approved by an ethics committee (IR.UMSHA.REC.1399.697), and informed consent was obtained, meeting ethical standards.

Limitations:

• The sample size (14 and 6 experts) is small, which the authors acknowledge as a limitation. This raises concerns about the robustness of criteria weights and system evaluations.

• The cross-sectional design limits insights into long-term HIS performance.

• The questionnaire’s development is described, but its full content or validation process (e.g., pilot testing, reliability analysis) is not detailed, which could affect the rigor of data collection.

Recommendations:

1. Provide more detail on the questionnaire’s development, including any validation steps or reliability measures (e.g., Cronbach’s alpha).

2. Justify the choice of the eight criteria and explain why other potentially relevant criteria (e.g., data security, interoperability) were excluded.

3. Consider sensitivity analysis to test the robustness of TOPSIS rankings against variations in criteria weights, addressing the small sample size concern.

Substantial Evidence for Conclusions

The study provides sufficient evidence for its conclusions:

• Criteria Weights: Table 1 and Table 2 show that cost and output quality received the highest weights (0.133 each), supported by expert scores. The process for calculating weights is transparent.

• HIS Rankings: The TOPSIS analysis (Tables 3–7) clearly demonstrates PARDAZESHGARAN’s superior performance (Cl = 0.959), followed by SAYAN (0.516) and RAYAVARAN (0.474). The decision matrix and weighted calculations are well-documented.

• Alignment with Literature: The discussion compares findings with prior studies, reinforcing the importance of cost, output quality, and support quality.

However, the small expert sample size and limited hospital scope weaken the generalizability of the rankings. Additionally, the manuscript notes discrepancies with Meidani et al. (2013), where RAYAVARAN ranked higher, but does not fully explore potential reasons beyond methodological differences (e.g., system updates, hospital-specific factors).

Recommendation: Discuss potential reasons for discrepancies with prior studies, such as changes in HIS versions or contextual factors. Acknowledge the limitations of the small sample size more explicitly in the conclusion and suggest how future studies could validate the findings.

Utility and Accessibility

The study’s utility lies in its systematic approach to HIS evaluation, which can guide hospital administrators in selecting systems. The TOPSIS method is accessible to those familiar with MCDM, and the use of Excel makes it feasible for resource-limited settings. However, the manuscript does not discuss the practical implementation of the findings (e.g., how hospitals can adopt PARDAZESHGARAN or improve SAYAN/RAYAVARAN). Additionally, the criteria weights and rankings are context-specific, and the manuscript could better address how other institutions could adapt the methodology.

Recommendation: Include a section or paragraph on practical applications, such as how hospitals can use the TOPSIS framework or prioritize criteria like cost and output quality. Provide guidance on adapting the methodology for different contexts.

Open Science Standards

• Data Availability: The manuscript states that data are available upon the journal’s request, which partially meets PLOS’s policy of full data availability. However, the data are not deposited in a public repository, which limits transparency.

• Software Availability: The study used Excel for TOPSIS calculations, which is widely available, but no custom scripts or templates are shared.

• Reporting Guidelines: The study does not explicitly mention adherence to reporting guidelines (e.g., STROBE for cross-sectional studies), though it follows a structured format.

• Reproducibility: The methodology is detailed enough to allow reproduction, but sharing the questionnaire and raw data would enhance this.

Recommendations:

1. Deposit the anonymized dataset (e.g., expert scores, decision matrices) in a public repository (e.g., Figshare) and provide an accession number.

2. Share the Excel template or TOPSIS calculation steps as supplementary material to aid replication.

3. Confirm adherence to relevant guidelines (e.g., STROBE) or explain why they were not applicable.

3. Additional Considerations

Literature Context and Fairness

The manuscript cites relevant studies (e.g., Farzandipour et al., 2017; Ker et al., 2018) to contextualize its findings and criteria. The literature review is comprehensive, covering HIS evaluation, TOPSIS applications, and related challenges. However, the discussion could better integrate these references to highlight how this study builds on or diverges from prior work. The treatment of the literature appears fair, with no evident bias.

Recommendation: Strengthen the discussion by synthesizing how this study’s findings align with or challenge specific prior studies, particularly regarding criteria prioritization and TOPSIS outcomes.

Clarity and Organization

The manuscript is well-organized, with clear sections (Introduction, Materials and Methods, Results, Discussion, Conclusion). The writing is generally clear, though some sections (e.g., TOPSIS steps) could be more concise. Tables are well-presented and support the narrative. The abstract and author summary effectively summarize the study, though the abstract could mention the small sample size limitation.

Recommendations:

1. Streamline the TOPSIS methodology section by referring readers to cited references for standard steps, focusing on study-specific adaptations.

2. In the abstract, briefly note the sample size limitation to set realistic expectations.

Potential for Revision

The study shows strong potential but would benefit from revisions to address the small sample size, data availability, and generalizability. The core methodology and findings are sound, and with enhancements, the manuscript could be suitable for publication.

Recommendation: Encourage the authors to revise and resubmit, focusing on the recommendations below.

4. Specific Comments for Authors

Strengths

• The use of TOPSIS provides a robust, systematic approach to HIS evaluation, which is valuable for decision-makers.

• The study is well-structured, with clear documentation of methods, results, and tables.

• The focus on cost and output quality aligns with practical priorities in healthcare settings.

• Ethical approval and expert panel selection enhance credibility.

Major Revisions

1. Sample Size and Generalizability: Acknowledge the small expert sample size (14 and 6) more prominently in the abstract and conclusion. Conduct a sensitivity analysis to test the robustness of TOPSIS rankings or discuss how a larger sample could validate the findings.

2. Data Availability: Deposit the anonymized dataset in a public repository and provide an accession number to meet PLOS’s open science policy. Share the Excel template or TOPSIS steps as supplementary material.

3. Criteria Justification: Clarify why the eight criteria were chosen and why others (e.g., security, interoperability) were excluded. Detail the questionnaire’s validation process.

4. Practical Implications: Add a section on how hospitals can apply the findings (e.g., adopting PARDAZESHGARAN, improving SAYAN/RAYAVARAN) and adapt the methodology for other contexts.

5. Literature Integration: Strengthen the discussion by comparing findings with specific prior studies and explaining discrepancies (e.g., Meidani et al., 2013).

Minor Revisions

1. Abstract: Mention the sample size limitation briefly.

2. TOPSIS Section: Condense standard TOPSIS steps and focus on study-specific details.

3. Reporting Guidelines: Confirm adherence to STROBE or explain its inapplicability.

4. Clarity: Proofread for minor grammatical errors (e.g., “Data Avallability” on Page 4) and ensure consistency in terminology (e.g., “Communication Quality Between Departments” vs. “interdepartmental communication quality”).**Reviewers' Comments:** Reviewer's Responses to Questions

**Comments to the Author**

1. Does this manuscript meet PLOS Digital Health’s publication criteria? Is the manuscript technically sound, and do the data support the conclusions? The manuscript must describe methodologically and ethically rigorous research with conclusions that are appropriately drawn based on the data presented.

Reviewer #1: Yes

Reviewer #2: Yes

2. Has the statistical analysis been performed appropriately and rigorously?

Reviewer #1: Yes

Reviewer #2: Yes

3. Have the authors made all data underlying the findings in their manuscript fully available (please refer to the Data Availability Statement at the start of the manuscript PDF file)?

Reviewer #1: No

Reviewer #2: No

4. Is the manuscript presented in an intelligible fashion and written in standard English?

Reviewer #1: Yes

Reviewer #2: Yes

5. Review Comments to the Author

Reviewer #1: The manuscript is a valuable contribution to HIS evaluation in digital health, with a sound methodology and relevant findings. However, it requires major revisions to address the small sample size, data availability, and generalizability, as well as minor revisions for clarity and reporting standards. The study has strong potential for publication in PLOS Digital Health if the attached issues are addressed.

Reviewer #2: 

Overall Assessment

The manuscript is a Multi-Criteria Approach Using TOPSIS in Hamadan, Iran" (PDIG-D-25-00100) submitted to PLOS Digital Health. As a reviewer, I find the manuscript to be a valuable contribution to the field of health informatics, addressing a critical need for systematic evaluation of Hospital Information Systems (HIS) in the context of Iranian teaching hospitals. The use of the TOPSIS method to rank HIS based on multi-criteria decision-making is novel and relevant, aligning well with the journal’s focus on advancing digital health technologies. However, the manuscript has several significant issues that require substantial revisions to meet the standards of PLOS Digital Health. Based on the importance of the topic and the potential for improvement, I recommend acceptance with major revisions. Below, I outline the key strengths, issues, and specific revisions needed to ensure the manuscript’s suitability for publication.

The manuscript addresses a timely and relevant topic by evaluating HIS, a cornerstone of digital health, using a rigorous multi-criteria decision-making approach (TOPSIS). Its focus on Iranian teaching hospitals fills a gap in the literature, where systematic HIS evaluation studies are limited. The study’s methodology, including the use of expert panels and literature-based criteria (e.g., cost, output quality, support quality), is well-grounded and demonstrates scientific rigor. The clear structure, academic tone, and alignment with ethical standards (e.g., ethical approval, informed consent) further support its potential for publication. The findings, particularly the ranking of PARDAZESHGARAN as the top HIS and the emphasis on cost and output quality, provide actionable insights for healthcare administrators.

Major Issues

Despite its strengths, the manuscript has several critical issues that must be addressed to ensure scientific accuracy, transparency, and impact. These issues necessitate major revisions, as they involve methodological corrections, enhanced clarity, and deeper interpretation.

Methodological Errors in TOPSIS Analysis:

o A critical error exists in Table 5 (p. 15, p. 33), where the positive and negative ideal solutions for the cost criterion are reversed. As cost is a negative criterion (lower is better), the positive ideal should be the lowest value (0.075), and the negative ideal should be the highest (0.079). This error likely affects the distances (Table 6) and relative closeness (Table 7), potentially altering the final rankings. This undermines the study’s validity and requires recalculation.

o The TOPSIS methodology description in the methods section is incomplete, covering only normalization and weighted matrix steps. Key steps, such as calculating distances to ideal solutions and relative closeness (CI), are omitted, reducing reproducibility.

Limited Methodological Transparency:

o The questionnaire development process lacks detail on the literature review or frameworks used (e.g., DeLone and McLean model, ISO standards), limiting transparency.

o The small sample sizes (14 experts for criteria weighting, 6 for HIS evaluation, and three hospitals) are acknowledged as a limitation but not justified. A rationale citing practical constraints or similar studies is needed.

o The ethical considerations section is brief, lacking specifics on consent procedures (e.g., written or verbal) and data protection measures (e.g., anonymization protocols).

Clarity and Presentation Issues:

o The results section lacks a narrative explanation of the TOPSIS process, making it difficult for readers unfamiliar with MCDM to follow how scores lead to rankings.

o An incorrect reference to Table 9 (p. 17, line 287) instead of Table 7 in the discussion risks confusing readers and indicates a need for careful proofreading.

o The absence of figures (e.g., a bar chart or radar plot comparing HIS scores) limits the visual communication of results, which is a missed opportunity for a journal emphasizing clear data presentation.

Weaknesses in Discussion and Conclusion:

o The discussion overinterprets small differences in criteria weights (e.g., technical quality: 0.112 vs. user satisfaction: 0.121) as evidence that users prioritize practical outcomes, requiring a more nuanced interpretation.

o The discussion does not sufficiently explore discrepancies with prior studies or discuss practical implications for stakeholders, limiting the study’s impact.

o The conclusion is brief and does not articulate the broader applicability of the findings or include a call to action for stakeholders.

Minor Issues

Inconsistent Terminology:

o The manuscript uses “multi-criteria approach” in the title and abstract but occasionally refers to “multi-criteria decision-making” elsewhere (e.g., introduction, p. 7). Consistent use of “multi-criteria decision-making” (MCDM), the standard term in operations research, would improve professionalism.

o The HIS name “Rayavaran” is inconsistently capitalized (e.g., “RAYAVARAN” in Table 3, p. 32, vs. “Rayavaran” in the text, p. 16). This should be standardized throughout.

Formatting Inconsistencies in Tables:

o The decimal places in tables are inconsistent. For example, Table 2 (p. 31) reports weights to three decimal places (e.g., 0.133), while Table 5 (p. 33) uses three or four decimal places (e.g., 0.075 vs. 0.0792). All numerical values should use a consistent number of decimal places (e.g., three decimals throughout).

o Table titles and captions could be more descriptive. For instance, Table 1 (p. 31) is titled “Average Scores for Key HIS Criteria,” but it could specify “Based on Expert Panel Evaluation” to clarify the source.

Minor Proofreading Errors:

o There are minor grammatical errors, such as “hospitals lacking robust HIS may struggle to remain competitive” (p. 7, introduction), which could be rephrased for clarity (e.g., “Hospitals without robust HIS may face challenges in maintaining competitiveness”).

o The phrase “data are ready for delivery following the journal’s request” (p. 4) is awkward. A clearer phrasing would be “Data are available upon journal request” or “Data will be provided as per journal requirements.”

Lack of Footnotes in Tables:

o Tables 1–7 lack footnotes explaining key terms or methods, such as the 5-point Likert scale (Table 1), normalization process (Table 4), or ideal solutions (Table 5). Adding brief footnotes would enhance accessibility for readers unfamiliar with TOPSIS or the study’s methods.

Vague Wording in Abstract:

o The abstract’s mention of “sample size” as a limitation (p. 1) is vague. Specifying whether this refers to the number of hospitals, experts, or systems evaluated would improve clarity without requiring major revision.

o The term “expert consultation” in the abstract is imprecise. A minor clarification, such as “expert panel scoring,” would align with the methods section.

Underdeveloped Contextual Details:

o The introduction briefly mentions the Iranian healthcare context but lacks specific details, such as HIS adoption rates or challenges in teaching hospitals. Adding a sentence or two with relevant statistics or context would strengthen the rationale without requiring major restructuring.

o The demographic data of the expert panels (p. 12, p. 30) are presented but not integrated into the narrative. A minor addition explaining their relevance (e.g., “The panel’s expertise ensured robust criteria weighting”) would enhance clarity.

Limited Use of Abbreviations:

o The manuscript introduces “Hospital Information Systems (HIS)” but does not consistently use the abbreviation throughout (e.g., “hospital information systems” appears in full in the discussion, p. 17). Consistently using “HIS” after its first introduction would improve readability.

Minor Redundancy in Discussion:

o The discussion repeats the importance of cost and output quality (p. 17) without adding new insights in some sentences. Streamlining these points (e.g., combining redundant statements) would improve conciseness.

6. PLOS authors have the option to publish the peer review history of their article (what does this mean?). If published, this will include your full peer review and any attached files.

**Do you want your identity to be public for this peer review?** For information about this choice, including consent withdrawal, please see our Privacy Policy.

Reviewer #1: **Yes: **Farima Safari

Reviewer #2: **Yes: **Sara S Nabavizadeh

**Figure resubmission:**While revising your submission, please upload your figure files to the Preflight Analysis and Conversion Engine (PACE) digital diagnostic tool, https://pacev2.apexcovantage.com/. PACE helps ensure that figures meet PLOS requirements. To use PACE, you must first register as a user. Registration is free. Then, login and navigate to the UPLOAD tab, where you will find detailed instructions on how to use the tool. If you encounter any issues or have any questions when using PACE, please email PLOS at figures@plos.org. Please note that Supporting Information files do not need this step. If there are other versions of figure files still present in your submission file inventory at resubmission, please replace them with the PACE-processed versions.**Reproducibility:**To enhance the reproducibility of your results, we recommend that authors of applicable studies deposit laboratory protocols in protocols.io, where a protocol can be assigned its own identifier (DOI) such that it can be cited independently in the future. Additionally, PLOS ONE offers an option to publish peer-reviewed clinical study protocols. Read more information on sharing protocols at https://plos.org/protocols?utm_medium=editorial-email&utm_source=authorletters&utm_campaign=protocols

---

## [Decision Letter · Decision Letter 1]

28 Aug 2025

Ranking and Selecting Hospital Information Systems: A Multi-Criteria Decision Making Approach Using TOPSIS in Hamadan, Iran

PDIG-D-25-00100R1

Dear Dr. mohammadpour,

We're pleased to inform you that your manuscript has been judged scientifically suitable for publication and will be formally accepted for publication once it meets all outstanding technical requirements.

Within one week, you'll receive an e-mail detailing the required amendments. When these have been addressed, you'll receive a formal acceptance letter and your manuscript will be scheduled for publication.

An invoice for payment will follow shortly after the formal acceptance. To ensure an efficient process, please log into Editorial Manager at https://www.editorialmanager.com/pdig/ click the 'Update My Information' link at the top of the page, and double check that your user information is up-to-date. For billing related questions, please contact billing support at https://plos.my.site.com/s/.

Kind regards,

Ali Nabavizadeh

Guest Editor

PLOS Digital Health

Additional Editor Comments (optional):

Reviewers' comments:

Reviewer's Responses to Questions

**Comments to the Author**

1. If the authors have adequately addressed your comments raised in a previous round of review and you feel that this manuscript is now acceptable for publication, you may indicate that here to bypass the “Comments to the Author” section, enter your conflict of interest statement in the “Confidential to Editor” section, and submit your "Accept" recommendation.

Reviewer #1: All comments have been addressed

Reviewer #3: (No Response)

2. Does this manuscript meet PLOS Digital Health’s publication criteria? Is the manuscript technically sound, and do the data support the conclusions? The manuscript must describe methodologically and ethically rigorous research with conclusions that are appropriately drawn based on the data presented.

Reviewer #1: Yes

Reviewer #3: Partly

3. Has the statistical analysis been performed appropriately and rigorously?

Reviewer #1: Yes

Reviewer #3: Yes

4. Have the authors made all data underlying the findings in their manuscript fully available (please refer to the Data Availability Statement at the start of the manuscript PDF file)?

Reviewer #1: Yes

Reviewer #3: No

5. Is the manuscript presented in an intelligible fashion and written in standard English?

PLOS Digital Health does not copyedit accepted manuscripts, so the language in submitted articles must be clear, correct, and unambiguous. Any typographical or grammatical errors should be corrected at revision, so please note any specific errors here.

Reviewer #1: Yes

Reviewer #3: No

6. Review Comments to the Author

Please use the space provided to explain your answers to the questions above. You may also include additional comments for the author, including concerns about dual publication, research ethics, or publication ethics. (Please upload your review as an attachment if it exceeds 20,000 characters)

Reviewer #1: The authors have done an excellent job of revising their manuscript and have diligently addressed all of my previous concerns. The paper is substantially improved, particularly with the welcome addition of a public dataset , stronger justification for the chosen criteria, and validation of the questionnaire. The discussion is now more robust, effectively contextualizing the findings against prior literature and explaining discrepancies , and the new section on practical implications significantly enhances the study's impact. While the small expert sample size remains a core limitation, the authors now transparently acknowledge this throughout the manuscript and have thoughtfully discussed the need for sensitivity analysis in future work. Given the thorough and responsive nature of these revisions, the manuscript is now suitable for publication.

Reviewer #3: 1. The mathematical foundation is not correct. We do not make mathematical operations on ordinal data. 

Likert scale is ordinal data, we do not perform addition, subtraction, multiplication or division on them.

The common approach is to use fuzzy sets to handle this situation. 

Therefore, you need to repeat the process of weighting the criteria and ranking in a correct form.

2. The link of the data set is not working.

3. The equations must be numbered.

7. PLOS authors have the option to publish the peer review history of their article (what does this mean?). If published, this will include your full peer review and any attached files.

**Do you want your identity to be public for this peer review?** For information about this choice, including consent withdrawal, please see our Privacy Policy.

Reviewer #1: Yes: Farima Safari

Reviewer #3: No
